# Comparative Study of Microbial Diversity in Different Coastal Aquifers: Determining Factors

María del Carmen Vargas-García [1] , Fernando Sola [2,*] and Ángela Vallejos [2]

1  Unit of Microbiology, Department of Biology and Geology, CITE II-B, University of Almeria, Marine Campus of International Excellence CEIMAR, 04120 Almeria, Spain
2  Water Resources and Environmental Geology, University of Almeria, 04120 Almeria, Spain
*  Correspondence: fesola@ual.es

**Abstract:** Coastal aquifers have been extensively studied from the hydrodynamic and geochemical points of view, but there is still a significant gap in the knowledge of their microbial diversity. The bacterial communities of four coastal aquifers at different depths and salinities were studied in order to infer the anthropogenic and physico-chemical influences on groundwater microbiota. At the physico-chemical level, samples from different aquifers, but with similar salinities, are more similar than those taken within the same aquifer. The microbial community at the phylum level shows the dominance of *Proteobacteria*, *Firmicutes*, and *Actinobacteria*. Samples from the same aquifer, although having very different salinities, are more similar than samples with similar physico-chemical characteristics. Therefore, the taxa present in these media are resilient to environmental variations. The aquifer preserving the most pristine conditions harbors the lowest values of biodiversity, compared to those affected by anthropic activities. The incorporation of pollutants into the aquifer favors the development of a so-called "rare biosphere", consisting of a high number of taxa which represent a low percentage (<1%) of the total microbial community. The analysis of microbial biodiversity in a coastal aquifer could be used as an indicator of the degree of anthropic alteration.

**Keywords:** coastal aquifer; saltwater; metagenomics analysis; microbiome diversity; anthropogenic impact





## 1. Introduction

Coastal aquifers constitute a fundamental hydrological connection between continental and marine water bodies [1]. The volume of fresh groundwater discharge into the sea could be similar to that of riverine flux [2,3]. Furthermore, this groundwater carries ions and dissolved nutrients that fertilize the ocean. In fact, groundwater discharge is the main source of biogeochemical nutrients in the ocean, being three or four times greater than the nutrient fluxes into the ocean from rivers [3–5]. Such aquifers are also important as a strategic resource of freshwater in coastal areas, which are usually highly populated and suitable for agriculture [6].

Typically, a coastal aquifer is composed of three layers with different properties: freshwater, interface, and saltwater. This vertical salinity distribution obeys, on the one hand, the origin of each of these water types; on the other hand, it obeys their densities. The freshwater layer comes from the infiltration of superficial water that flows from the continent to the coast. This water usually has low salinity, but slightly increases its ionic content with the dissolution of the aquifer substratum during its movement from inland to sea. Conversely, the saltwater layer comes from the sea and has approximately the same salinity as seawater. It has a wedge shape that reduces its thickness landward until it disappears. The interface is a mixing zone, with intermediate characteristics between fresh- and saltwater [6,7].

Although coastal aquifers have been intensively studied on a global scale [6,8–10], less attention has been given to them from the microbiological point of view, despite their fundamental role in the fertilization process of the ocean [11–14].

In general, coastal aquifers are environments with pronounced gradients of physico-chemical parameters, particularly salinity, but also temperature, pH, eH, and dissolved oxygen shift from the top of the aquifer to the bottom, and along the flow direction from land to coast [15–17]. For that reason, this environment is an ecosystem where a high variety of specialist microorganisms can live [12,18–22]. The activity of the microbiota also modifies the environment, generating new opportunities for new organisms [23]. The study of microbial communities in coastal aquifers has increased in recent decades [14,21,22,24–28], not only because of the wide diversity they comprise, as already mentioned, but also because of the concurrence of a series of characteristics, such as the existence of transition zones, affected by a wide range of conditions or important anthropogenic pressure [6,29], which make them habitats of special interest from a microbiological perspective. Despite this, the existing knowledge of their microbial diversity and associated biogeochemical processes is very limited in comparison with continental and marine waters [14,21].

The interface is a strip of the aquifer where many chemical reactions take place, due to the mixing of waters with different characteristics [30–32]. Some of these reactions, such as metal oxidation/reduction, or denitrification, can be mediated by bacteria [12,33–39]. Moreover, microorganisms are postulated to be primarily responsible for most of the complex processes that take place in coastal aquifers, especially in the transition zone between groundwater and seawater [18,40]. Thus, the taxonomic diversity of the microbiome is complemented by its functional diversity, which translates into the presence of heterotrophs, chemolithotrophs, methanotrophs, ammonia oxidizers, nitrifying and denitrifying microorganisms, hydrogen oxidizers, sulfur oxidizers, phenol oxidizers, iron reducers, sulfate reducers, and propanotrophs [41,42].

In most cases, no single species is capable of carrying out processes that require multiple reactions, meaning that the joint action of several microorganisms is required [43]. The specific nature of each of these reactions, determined by nutritional, environmental, and physico-chemical conditions, will modulate the composition of the microbial associations mentioned. In this sense, the most versatile microorganisms, from a metabolic point of view, will be able to proliferate, or at least play a relatively important role in, different habitats. These are the so-called core species [44]. On the contrary, those that exhibit a more restricted or specialized metabolism will only be able to thrive in environments where the appropriate conditions are present. The latter microorganisms could act as environmental indicators [24,45], thus making it possible to establish the incidence of changes or disturbances in an aquifer's properties by relation to the presence of these microorganisms.

The Spanish Mediterranean coastline contains numerous aquifers with varying lithological characteristics. These aquifers have been studied intensely from the hydrodynamic and hydrogeochemical points of view [46–49]. Although the microbial dimension of coastal aquifers has been less explored than their physico-chemical dimension, in the case of coastal aquifers in Spain, we can find a great number of works on the microbiological content of their waters, both in detrital aquifers [21,22,50–53] and in carbonate aquifers [18,23].

Unlike previous studies, this paper does not focus on a single aquifer, but on four aquifers close to each other, in order to compare their microbiological composition and the possible factors that determine it: the nature of the aquifer substrate (detrital vs carbonate), the degree of anthropic influence, and the range of salinity. In each of the four different selected areas in the southeast coast of Spain, samples from the freshwater, interface, and saltwater layers were taken in order to study their chemical and microbiological content. Two of these areas are detrital, and the other two are carbonate. The type of substratum (detrital vs. carbonate), the physico-chemical parameters of groundwater, and the anthropic influence were studied in order to elucidate what factors govern microbial diversity in coastal aquifers.

## 2. Materials and Methods

### 2.1. Study Sites, Sampling, and Physico-Chemical Parameters

Four different areas at which to conduct this study were selected: Aguadulce, Palmer, the Andarax delta, and the Cabo de Gata littoral plain (Figure 1A). Two of these zones (Aguadulce and Palmer) are carbonate aquifers, and are composed essentially of limestone and dolostone. In contrast, the Andarax delta and the Cabo de Gata littoral plain are detrital aquifers, composed of layers of gravel, sand, silt, and clay.

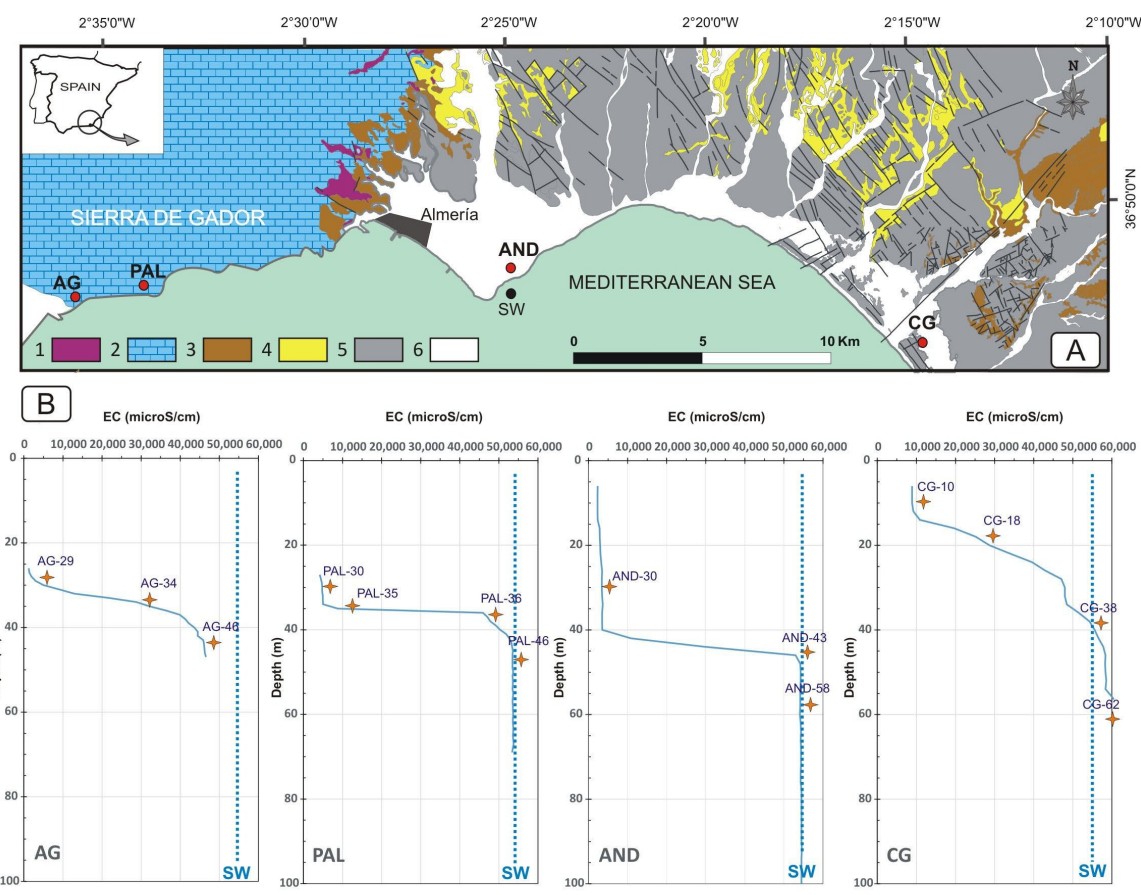

**Figure 1.** (**A**): Location of study sites (AG—Aguadulce, PAL—Palmer, AND—Andarax delta, CG—Cabo de Gata littoral plain), SW—seawater and geological context (1—metapelite, 2—carbonate, 3—coarse neogene sediments, 4—fine neogene sediments, 5—quaternary deposits, 6—holocene alluvial deposits). (**B**): Electrical conductivity (EC) profiles of groundwater (dotted line—seawater EC; star—sampling point depth).

Aguadulce and Palmer belong to the Sierra de Gador, a 40 km long anticline composed mainly of Triassic carbonate rocks [48]. The Sierra de Gador connects with the sea along a 7 km stretch of land, in which Aguadulce is the site of natural discharge for the aquifer, while in the Palmer area, the groundwater discharge into the sea is diffuse. The Andarax delta and the Cabo de Gata littoral plain belong to the Almería-Nijar Basin. It is a Neogene basin where the aquifer materials are composed of deltaic plioquaternary deposits. In the area where the boreholes used in this study are located, these materials have an approximate depth of 100 m. Three of the aquifers studied (Aguadulce, the Andarax delta and the Cabo de Gata coastal plain) are located below areas of intense farming activity [54–57], well-known as main intensive agricultural areas of Europe [58,59]. On the other hand, the Palmer aquifer can be considered a pristine aquifer, as it is in a mountainous area with no anthropic activity.

PVC (polyvinyl chloride) well screens were used in the entire formation for all boreholes, which were drilled specifically for the purpose of carrying out this research, with sampling taking place more than 6 months after drilling, so that the drilling work would not interfere with the bacterial communities present in the groundwater.

A sampling campaign was carried out in September 2021. The sampling point in Aguadulce corresponded to a 50 m deep borehole 100 m from the sea. In this borehole (AG), samples were taken at 29, 34, and 46 m depths (AG-29, AG-34, and AG-46), corresponding to freshwater, mixing, and saltwater zones, respectively. As for the Palmer aquifer, a 70 m deep borehole 150 m from the coastline was selected (PAL). In this case, four samples were taken at 30, 35, 36, and 46 m depths (PAL-30, PAL-35, PAL-36, and PAL-46). In the Andarax delta, the sampling point was 100 m in depth, located 260 m from the coast, and the samples were taken at 30, 43, and 58 m depths (AND-30, AND-43, and AND-58). The Cabo de Gata aquifer was sampled thanks to a borehole located 600 m from the sea. This borehole was 76 m in depth and the samples for this study correspond to 10, 18, 38, and 62 m depths (CG-10, CG-18, CG-38, and CG-62). A seawater sample (SW) was also taken from the coast next to the study area.

Temperature, electrical conductivity (EC), concentration of dissolved oxygen, and pH were determined in situ. Alkalinity (as $HCO_3$) was determined by titration at the time of sampling. The multi-level sampling technique was used to collect groundwater samples from the top to the bottom of the boreholes, using a discrete-interval sampler (Solinst Mod. 425). Sample concentrations of major anions and cations (Cl, $SO_4$, Ca, Mg, Na, K and $NO_3$) were measured using standard methods, such as ICP-Mass Spectrometer, Ion Chromatography, and X-ray fluorescence (Scientific Instrumentation Centre, University of Almeria, Almería, Spain). The analytical results obtained are summarized in Table 1.

**Table 1.** Results of chemical analyses of water samples in meq/L. Location of sampling points in Figure 1A. (SW—seawater, EC—electrical conductivity, %SW—seawater percentage, W-TYPE—hydrochemical facies/water type).

| SAMPLE | pH | EC (µS/cm) | O$_2$ (mg/L) | T (°C) | Ca (meq/L) | Mg (meq/L) | Na (meq/L) | K (meq/L) | Cl (meq/L) | HCO$_3$ (meq/L) | SO$_4$ (meq/L) | NO$_3$ (mg/L) | DOC (mg/L) | %SW | W-TYPE |
|---|---|---|---|---|---|---|---|---|---|---|---|---|---|---|---|
| AND-30 | 7.14 | 3660 | 6.99 | 21.5 | 12.57 | 10.94 | 18.78 | 0.61 | 15.26 | 5.90 | 17.11 | 100.00 | 0.0004 | 2 | Na-Ca-Mg-SO$_4$-Cl |
| AND-43 | 7.48 | 27,600 | 8.14 | 21.5 | 20.11 | 48.36 | 267.69 | 5.24 | 260.65 | 5.40 | 34.17 | 95.58 | 0.0026 | 45 | Na-Cl |
| AND-58 | 7.45 | 55,600 | 5.5 | 21.6 | 27.74 | 93.17 | 506.55 | 10.87 | 561.64 | 3.90 | 54.78 | 3.45 | 0.0005 | 98 | Na-Cl |
| AG-29 | 7.70 | 3970 | 8.16 | 21.3 | 5.14 | 7.15 | 31.88 | 0.61 | 29.90 | 3.90 | 4.93 | 30.28 | 0.0000 | 5 | Na-Cl |
| AG-34 | 7.84 | 29,800 | 6.75 | 21.2 | 16.22 | 53.21 | 274.67 | 5.65 | 293.37 | 3.60 | 29.98 | 30.88 | 0.0008 | 51 | Na-Cl |
| AG-46 | 7.73 | 47,800 | 8.16 | 20.8 | 21.31 | 86.60 | 419.65 | 9.16 | 480.96 | 2.70 | 47.60 | 4.21 | 0.0028 | 84 | Na-Cl |
| CG-10 | 7.35 | 9330 | 5.21 | 22.7 | 11.33 | 17.52 | 61.14 | 1.10 | 66.57 | 6.00 | 19.49 | 216.62 | 0.0026 | 11 | Na-Cl-SO$_4$ |
| CG-18 | 7.47 | 25,600 | 6.68 | 22.3 | 16.92 | 39.56 | 225.76 | 4.27 | 233.85 | 6.40 | 37.16 | 21.57 | 0.0031 | 41 | Na-Cl |
| CG-38 | 7.31 | 57,400 | 6.01 | 22.7 | 32.98 | 102.22 | 537.12 | 10.90 | 592.95 | 4.20 | 60.15 | 0.94 | 0.0061 | 104 | Na-Cl |
| CG-62 | 7.05 | 65,700 | 4.58 | 23.3 | 37.72 | 109.62 | 606.99 | 12.51 | 691.11 | 4.20 | 68.71 | 0.51 | 0.0044 | 121 | Na-Cl |
| PAL-30 | 8.00 | 5040 | 7.92 | 22.4 | 5.24 | 10.77 | 42.79 | 0.66 | 38.93 | 3.80 | 4.93 | 12.33 | 0.0024 | 7 | Na-Mg-Cl |
| PAL-35 | 7.71 | 15,770 | 8.26 | 22.2 | 14.67 | 49.01 | 119.65 | 4.07 | 143.30 | 3.50 | 14.80 | 10.55 | 0.0013 | 25 | Na-Mg-Cl |
| PAL-36 | 7.40 | 48,400 | 8.61 | 22.2 | 28.99 | 110.28 | 432.31 | 10.26 | 486.32 | 3.10 | 48.84 | 1.64 | 0.0084 | 85 | Na-Cl |
| PAL-46 | 7.37 | 56,100 | 4.52 | 21.5 | 26.30 | 107.81 | 497.82 | 13.45 | 567.56 | 3.00 | 56.90 | 0.77 | 0.0014 | 99 | Na-Cl |
| SW | 7.99 | 54,500 | — | 16.4 | 28.89 | 106.33 | 510.92 | 11.33 | 572.07 | 2.90 | 58.65 | 0.00 | 0.0039 | 100 | Na-Cl |

Groundwater samples were also taken at different depths in order to study their prokaryotic communities, given that the screened area spans the entire piezometer. Depths where the groundwater was sampled correspond to freshwater, freshwater-saltwater interface, and saltwater. These depths were previously determined by means of vertical logs using a temperature-conductivity multimeter (Solinst 107 TLC meter, Solinst Canada Ltd., Georgetown, ON, Canada). The samples were taken using a stainless steel sampler without pumping, to prevent vertical mixing of groundwater, and the sampler was sterilized after each use. For the analysis of microbiology, biomass from one liter of water from each depth was stored in sterilized polyethylene bottles at 4 °C. Once in the lab, they were filtered using 0.22 µm mixed cellulose ester filters (CAT. NO: GSWG047S6, Millipore, Burlington, MA, USA), and the filters were frozen at −20 °C until analysis.

*2.2. DNA Extraction and Sequencing Analysis*

The filters containing microbial cells from different aquifers and depths were sliced and placed into tubes in order to achieve DNA extraction through the application of the QIAsymphony PowerFecal Pro DNA kit (Qiagen, Hilden, Germany). Cell lysate was performed using mechanical disruption and enzyme treatment. Subsequently, the DNA was purified from the sample on a silica/gel column that allowed DNA isolation and the cleaning of contaminants and inhibitors of future reactions. Thereafter, the quality and concentration of the DNA were evaluated, using Nanodrop 2000 (Thermo Fisher Scientific, Waltham, MA, USA), and used to construct the corresponding genomic libraries from the V3-V4 hypervariable region of the 16S rRNA gene [60], using universal primers S-D-Bact-0341-b-S-17 (5′-CCTACGGGNGGCWGCAG-3′) and S-D-Bact-0785-a-A–21 (5′-GACTACHVGGGTATCTAATCC-3′) and the following PCR conditions: initial denaturation at 95 °C for 5 min; 25 cycles of amplification consisting of denaturation at 95 °C for 40 s, annealing at 55 °C for 2 min, and extension at 72 °C for 1 min; and final extension at 72 °C for 7 min. Sequencing analysis was carried out on the Illumina MiSeq platform (Illumina, Inc., San Diego, CA, USA) following a 300 bp × 2 paired-end protocol. Overlapped sequences were constructed using the BBMerge module from BBMap software V.38, considering a minimum overlap of 70 nts at each end, and CUTADAPT software V.1.8.1 was utilized for the removal of adapters [61]. Module BBReformat was used to select sequences showing quality scores (Q) higher than Q20 and lengths > 200 bp, while Module CD-HIT-dup of the CD-HIT V.4.8.1 allowed the removal of chimeric sequences [62]. The same CD-HIT software was applied to group sequences showing at least 99% similarity [63]. The representative sequences' OTUs (Operational Taxonomic Units) were taxonomically assigned using a combination of two strategies in order to obtain a more precise and comprehensive identification. Each sequence was compared against the NCBI (National Center for Biotechnology Information) nt database using the BLAST (Basic Local Alignment Search Tool) strategy of local alignment [64] to associate each of the OTUs with one of the taxonomic groups in the database. Taxonomies assigned with less than 97% identity were reassigned with the one obtained by the NBAYES (Naïve Bayesian) strategy [65] with the 16S-specific SILVA v.138 database. The sequence data were deposited in the NCBI SRA under the BioProject ID PRJNA894943.

*2.3. Statistical Analysis*

Diversity indices (Shannon–Wiener, Chao 1, and Simpson), as well as OTUs, were estimated from the microbiome taxonomic profiling service at the EzBioCloud website (http://www.ezbiocloud.net/contents/16smtp (accessed on 31 May 2022)). SIMPER (similarity percentage analysis) and indicator species analysis were performed using PAST software V.4.08 [66], as was the cluster analysis. The associations among variables were plotted through principal component analysis (PCA). All of the statistical treatments were performed at a 95% confidence level using IBM SPSS Statistics V.27 software (IBM Analytics, Armork, NY, USA) and Statgraphics Centurion V.19 (Statpoint Technologies, Inc., Warrenton, VA, USA).

## 3. Results

*3.1. Hydrochemical Characterization*

The water in the four coastal aquifers is predominantly characterized by sodium chloride facies, increasing in saline content with depth (Table 1). The Aguadulce aquifer (AG) is characterized by a slightly developed freshwater layer, which reaches no seawater values at any point in the entire profile studied (Figure 1B). As for the Palmer aquifer (PAL), it displays a profile that is more characteristic of a coastal aquifer, with clearly defined sections of freshwater, interface, and saltwater. The same occurs in the Andarax delta (AND), which features a highly developed freshwater section. Finally, the aquifer in the Cabo de Gata littoral plain (CG) is characterized by having the most saline waters in the

entire study, with values exceeding the salinity of seawater in its deepest parts, as it is evaporated marine paleowater [67,68].

The chemical composition of the studied waters indicates a wide variability in relation to the depth of the sample, but not so much regarding the aquifer considered (Figure 2A). Despite this variability, there is parallelism in the trends, with the exception of the shallowest sample from the Andarax delta aquifer, which is characterized by greater concentrations of calcium sulphate than the rest of the samples.

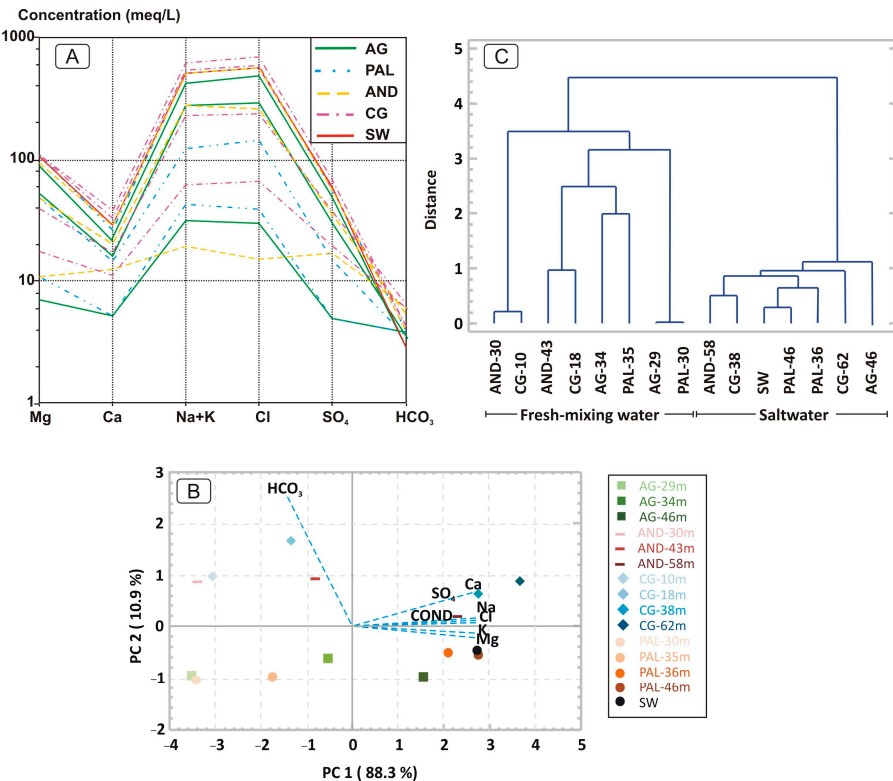

**Figure 2.** (**A**): Schöeller-Berkaloff diagram showing groundwater and seawater concentrations. (**B**): Principal component analysis of physico-chemical variables and observations. (**C**): Dendrogram of cluster analysis between observations corresponding to samples taken at different depths in the aquifers.

Based on the physico-chemical parameters measured and normalized for these data previously, the principal component analysis (Figure 2B) shows that the parameters linked to salinity/conductivity are related to component 1, while alkalinity (as $HCO_3$) is described as component 2. The samples taken at greater depths, independently of the aquifer considered, are grouped with component 1, which explains the higher salinity of this medium. The hierarchical cluster analysis of this study used Ward's method with Euclidean distance. Ward's method uses an analysis of variance approach in order to evaluate the distances between clusters, and minimizes the sum of the squares of distance between any two clusters that can be formed at each step of the cluster classification. This produces a dendrogram—a visual representation of the linkage distance during the history of cluster merging. The cluster analysis obtained corroborates the importance of the salinity of groundwater (Figure 2C). The deepest and saltiest samples, along with seawater, are grouped at a close distance, while the samples from the freshwater and interface layers constitute part of another clearly distanced cluster. A differentiation can be observed within both clusters based on the type of aquifer, such that the samples taken from carbonate aquifers (AG and PAL) are distinct from those taken in the detrital aquifers (AND and CG).

### 3.2. Composition of the Microbial Community

The microbial community's structure at the taxonomic levels of phylum (Figure 3A) and genus (Figure 3B) is shown in relation to each of the aquifers analyzed, and at the different depths studied. Regarding the level of phylum, the results show the dominance of three different phyla: *Proteobacteria*, *Firmicutes*, and *Actinobacteria*. *Proteobacteria* ranged in relative abundance from 16% to 95% and was the dominant group in the PAL aquifer, where relative abundances were always above 85%, and in the seawater sample, which reached values of approximately 73%. In contrast, their representation was lower in the communities associated with the AG aquifer (16–44%) and, above all, those associated with the CG aquifer (21–30%). The high presence of *Proteobacteria* in the SW and PAL samples left little room for other microbial groups. *Firmicutes* and *Actinobacteria* were the other two most dominant phyla, although their relative abundances were always lower than 4%, except in the samples corresponding to the shallowest layers of the PAL aquifer, where the presence of *Firmicutes* was quantified at 6.86% and 4.57% for 30 m and 35 m, respectively. In contrast, the community associated with the CG aquifer was dominated in all cases by species belonging the phylum *Firmicutes*, especially in the shallowest samples, corresponding to 10 m and 18 m depths (68.20% and 48.23%, respectively). This pattern was also repeated in the case of the AG samples, where the relative abundance of *Firmicutes* at the shallowest depth (29 m) was maximal (48.68%), although their presence at greater depths decreased drastically. The microbiome of this aquifer at greater depths was dominated by representatives of *Actinobacteria* (59.20% and 28.20% at 34 m and 46 m, respectively) and, as mentioned above, *Proteobacteria* (44.10% at 46 m). Finally, members of the phylum *Bacteroidetes* reached significant levels in the communities associated with the SW samples (16.13%) and at specific depths of the different aquifers: 7.38% in PAL-46, 5.73% in CG-38, 8.96% in AND-43, and 4.38% in AG-34.

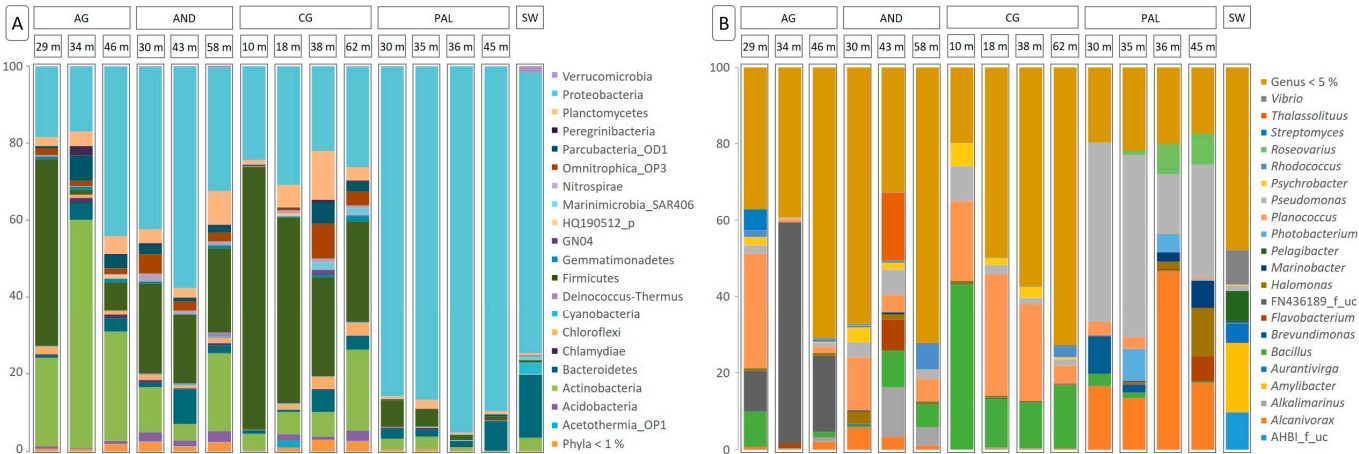

**Figure 3.** Distribution of relative abundances of the dominant microbial groups in the different aquifers and depths studied. (**A**): Phylum taxonomic level (representatives with relative abundance > 1% in at least one of the samples). (**B**): Genus taxonomic level (representatives with relative abundance > 5% in at least one of the samples).

The importance of these phyla was confirmed by SIMPER analysis (Figure 4A), according to which these three microbial groups contributed more than 60% to the differences between the microbial communities from the different aquifers. In this regard, the role assigned to the phylum *Acetothermia_OP1*, with a contribution of 27.88% to these differences, should be highlighted because of its exceptional presence in the microbiome of sample CG-18m and its virtual absence in all other cases. In addition, according to the results generated by the indicator species analysis, other phyla could be highlighted that, due to their abundance, marked noticeable differences between aquifers. This is the case of *Chlamydiae*,

*Gemmatomonadales,* and *Parcubacteria_OD1* for AG; *Acidobacteria, Deinococcus-Thermus,* and *Nitrospirae* for AND; *Chloroflexi* for CG; and *Bacteroidetes* for SW.

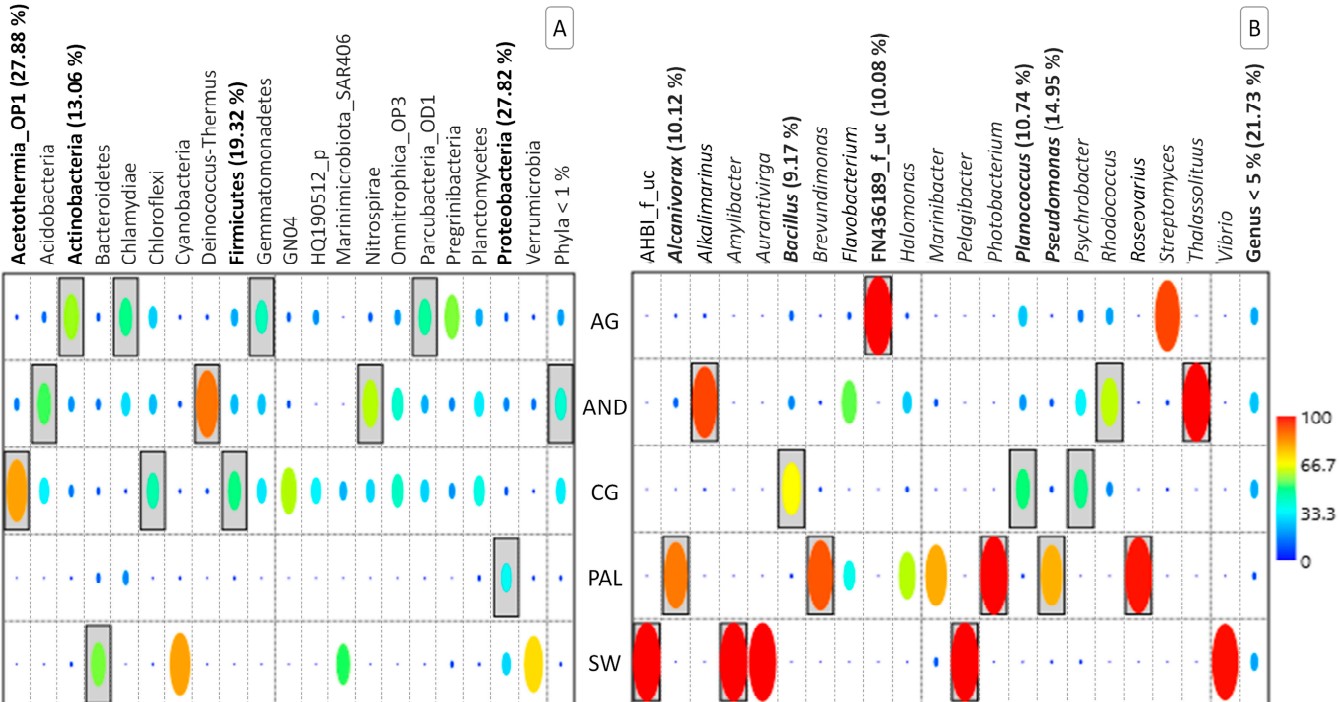

**Figure 4.** Species indicator analysis at the taxonomic levels of Phylum (**A**) and Genus (**B**). Statistically significant taxonomic groups are boxed, while the names of those that, according to SIMPER analysis, contributed most to establishing the overall differences between aquifers, are highlighted in bold. The percentage contribution explaining the observed differences in each case is included in parentheses.

At the genus taxonomic level, unlike what was observed in the case of the phylum level, it was not possible to establish one or several globally dominant genera (Figure 3B). On the contrary, each aquifer was characterized by showing a microbiome in which a different microbial group prevailed, with the existence of exceptional similarities. Thus, in the AG aquifer, the most abundant genus was *FN436189_f_uc* (10–58%), with high levels of *Planococcus* and *Bacillus*, 30.09% and 9.15%, respectively, in the shallowest sample. These last two genera were also dominant in the microbial community associated with the CG aquifer (4–31% and 11–43%, respectively), with a relatively dominant presence of *Pseudomonas* (9.15%) and *Psychrobacter* (6.02%) at the 10 m depth. *Pseudomonas* was also abundant in the samples from the PAL aquifer (15–48%), accompanied in this case by *Alcanivorax* (13–46%). In the last aquifer studied—AND—the microbiomes found at the different depths analyzed were characterized by the absence of a dominant genus. *Planococcus* was the genus with the highest relative abundance at 30 m (13.72%), while *Thalassolituus* was most abundant at 43 m (17.81%). At 48 m, no genus stood out. In the case of seawater, the most abundant genus was *Amylibacter* (17.83%). A common feature in most cases was the high proportion of genera with relative abundances below 5%, except in the PAL samples. There was also a general increase in this proportion as depth increased.

SIMPER analysis at this taxonomic level highlighted the relevance of many of the genera cited in establishing structural differences between aquifers (Figure 4B). *Pseudomonas*, *Planococcus*, *Alcanivorax*, *FN436189_f_uc*, and *Bacillus* were the main contributors to this variability, which, together with the Phyla < 1%, which made the highest contribution, accumulated 76.79% of significance. Phyla < 1% refers to those phyla that, although perfectly identified, show relative abundance levels below 1%. In the different analyses carried out to determine the statistical significance of the different microbial groups, only those

that exceed a certain level of abundance (in this case, 1%) are considered. The remaining microorganisms, representing less abundant phyla, are included under this common heading. Other genera significantly highlighted in each of the aquifers, according to the results obtained by applying the indicator species analysis, were as follows: *Alkalimarinus*, *Rhodococcus*, and *Thalassolituus* for AG samples; *Brevundimonas*, *Photobacterium*, and *Roseovarius* for PAL; and *AHBI_f_uc*, *Amylibacter* and *Pelagibacter* for SW.

### 3.3. Microbial Diversity

The Shannon, Chao1, and Simpson indices, as well as observed OTUs (Figure 5), were used to determine the α diversity of the samples. In terms of richness, the average number of OTUs and the Chao 1 estimator showed similar patterns, with higher values associated with AND samples (in the range of 2000–3000) and lower ones (between 500 and 1000) associated with PAL samples. Regarding the Shannon diversity index, measuring both richness and evenness, in most cases the values obtained were higher than 4. Only in the case of the samples from the PAL aquifer, in addition to two others from the AG and CG aquifers, at intermediate and minimum depths, respectively, the diversity associated with the microbial community was less than 4. Finally, regarding the uniformity of the samples determined through Simpson's index, the values found showed highly heterogeneous communities (above 0.875) [69]. Only the sample representative of the intermediate depth of the AG aquifer (AG-34m) fell below this value, although the level found (close to 0.75) can also be considered typical of a highly heterogeneous community. The results found for the SW sample in relation to these last two indices represent microbial communities of intermediate diversity between the extreme values generated by the microbiomes of the AND and PAL aquifers.

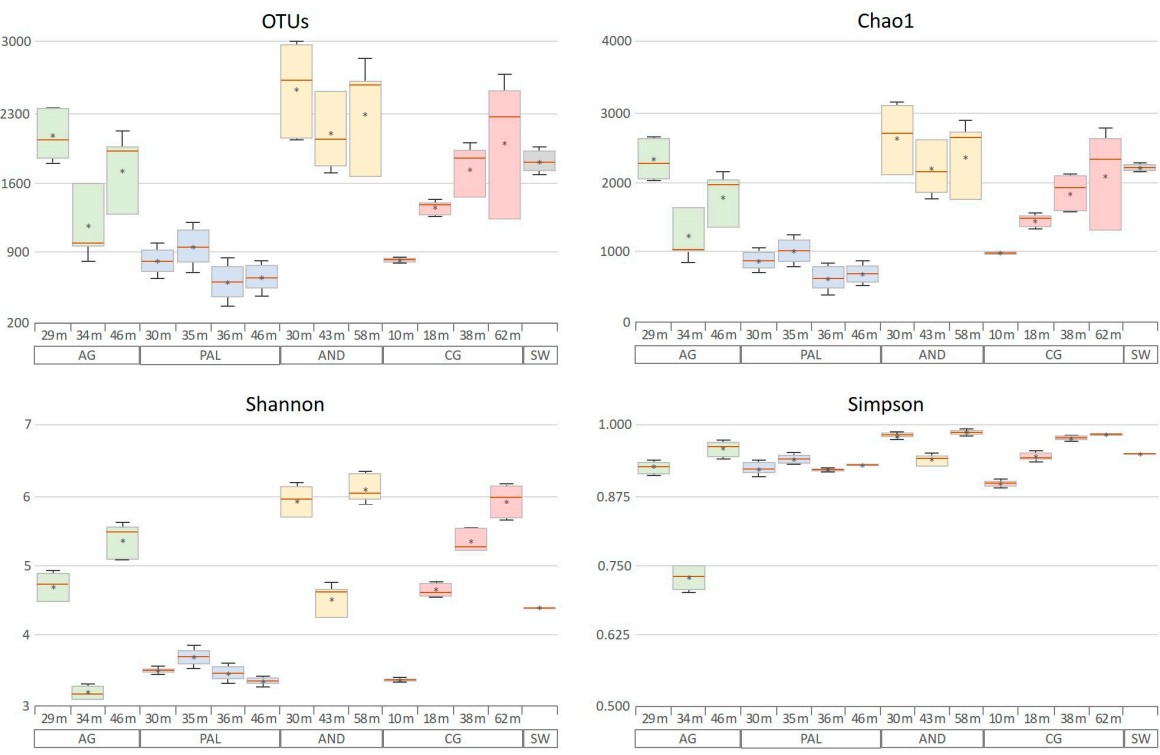

**Figure 5.** Alpha diversity values found for the different aquifers and depths studied. The red line indicates the median value and the asterisk indicates the mean. The lower limit of the box marks the position of the first quartile and the upper limit the position of the third quartile. The upper and lower whiskers represent the maximum and minimum values, respectively.

No clear pattern was observed for the α diversity indices due to variation in sampling depth. Thus, in the AG and AND aquifers, intermediate depths always resulted in minimum diversity values. In contrast, in the PAL aquifer, no significant variations in diversity indices were detected as depth changed, whereas in the case of the CG aquifer, a clear increase in diversity could be observed as depth increased.

β diversity was studied through principal component and cluster analysis (Figure 6). According to the results provided by the latter, the PAL aquifer showed a sufficient degree of homology in its microbiome, regardless of depth level. On the contrary, in the case of the AG and CG aquifers, depth marked clear differences in the microbial community. Thus, the samples corresponding to the shallowest depths of both aquifers were included in the same cluster, while those corresponding to the deepest depths constituted two clusters. The seawater is clearly separated from all aquifers, and it is related to the phyla *Cyanobacteria*, *Bacteroidetes*, *Verrumicrobia,* and *Proteobacteria*.

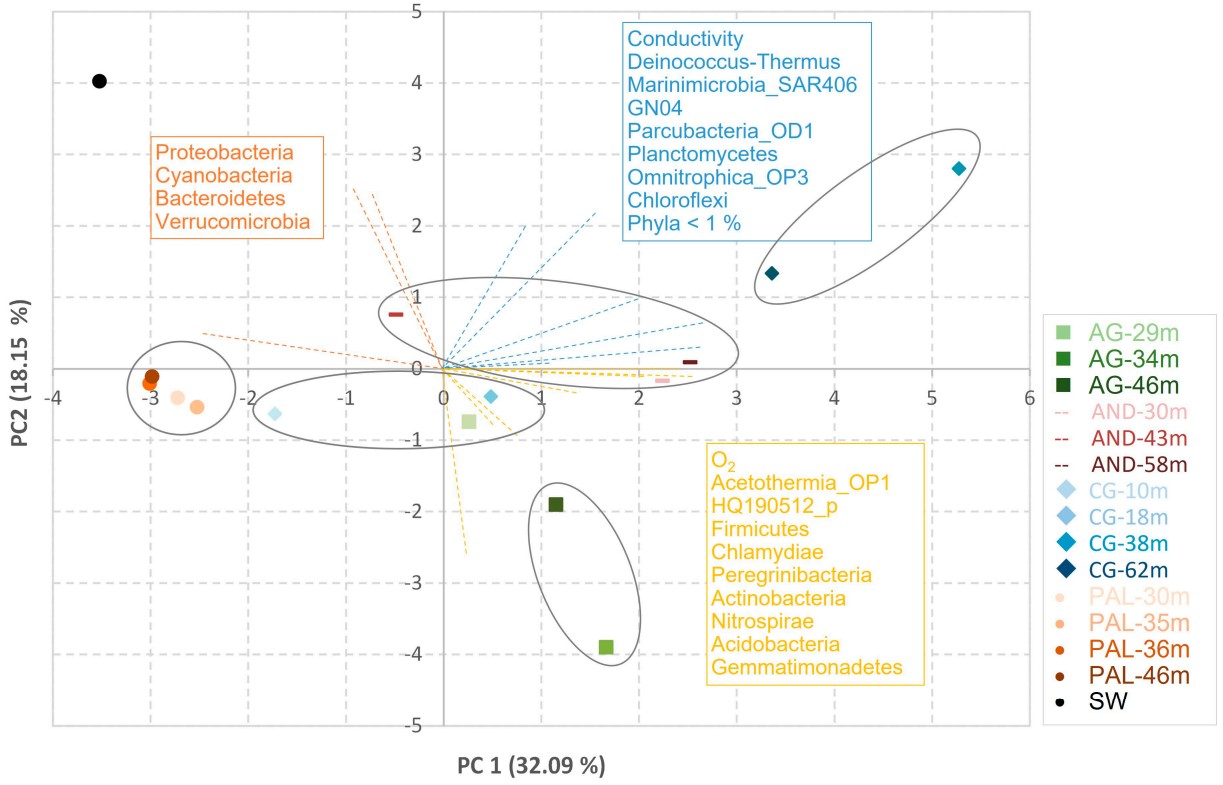

**Figure 6.** PCA-biplot analysis of microbial data at phylum taxonomic level. The nearest neighbor algorithm and square Euclidean distance were applied for hierarchical clustering of samples.

## 4. Discussion

In recent years, knowledge about the microbiota in coastal aquifers has increased, thanks, in part, to recent reviews [14,27]. Microbial communities associated with groundwater are characterized by relatively high diversity, in many cases with taxonomic affiliations and functionalities apparently far from those expected according to environmental conditions and the geological classification of the aquifer [70]. To some extent, this fact can be considered a consequence of the high adaptability shown by the species typical of this type of habitat, which are capable of resisting conditions that are not very favorable for life, as well as sudden alterations in these conditions [71]. Some members of the phyla *Proteobacteria*, *Firmicutes,* and *Actinobacteria*, dominant in the aquifers studied here (Figure 3A), fit the profile of metabolically versatile microorganisms, especially in the case of the first [18]. The *Proteobacteria* phylum comprises microorganisms of great metabolic plasticity, which allows them to be dominant in environments of highly varied characteristics, such as both

pristine and contaminated aquifers. In pristine aquifers, systems that have received much less attention than those showing contamination [50], this phylum has been detected as the majority regardless of the mineralogical nature of the aquifer [72,73]. This is precisely the case of the PAL aquifer, a carbonate system located in an area with a limited impact from anthropic activity, as indicated by the low concentrations of nitrates measured in the water of this aquifer (Table 1). In cases such as this, only the soil microbiota can have any influence when there is a connection with the aquifer [44]. In pristine aquifers, the influence of geochemical characteristics on the shaping of the microbial community is greater than in contaminated environments [74–76]. In the latter, the identity and relative abundance of microbial species is affected by the presence of compounds of anthropogenic origin [77].

In the second of the carbonate aquifers, AG, the presence of *Actinobacteria* stood out in comparison to the rest of the groundwater analyzed. However, the microbial profile of this aquifer showed a strong dependence on the depth analyzed, since the dominant phylum in each case was different: *Firmicutes* at 29 m, *Actinobacteria* at 34 m, and *Proteobacteria* at 46 m (Figure 3A). A higher presence of *Firmicutes* and *Actinobacteria* can be linked to a lower content of biodegradable dissolved organic carbon [78], an aspect that may be related to the greater capacity of the microorganisms assigned to these phyla to metabolize carbon substrates of greater complexity and recalcitrance [79]. The lowest dissolved organic carbon values are observed in the AG aquifer (Table 1). One such substrate example is humic acids, highly stable compounds with a high affinity for clay, with which they form complexes [80]. In aquifers, their presence is more common in shallow areas [81], which would support the high proportion of *Firmicutes* in the shallowest samples, not only in the case of the AG aquifer, but also in the CG aquifer. It should be remembered that the latter is a detrital aquifer, in which clay layers appear. A third phylum associated with the presence of more refractory substrates and lower biodegradable dissolved organic carbon is *Planctomycetes* [82], which is also beginning to be recognized by the ubiquity of its members, with particular relevance in aquatic environments [83]. Members assigned to this phylum were also detected in the AG, AND, and CG aquifers (Figure 3A), with their relative abundances generally increasing with sampling depth, as has been described in other aquifers [79]. The concentrations of dissolved organic carbon measured in the samples of theses aquifers (Table 1) are very low (<0.01 mg/L).

Ultimately, neither this factor, depth, nor the type of aquifer were shown to be decisive in defining the microbiome, as demonstrated by the principal component and cluster analyses. In specific cases, such as the PAL and AND aquifers, of different geological origin, the similarities found in the microbiomes associated with different depths of the same aquifer were high enough for all of the samples to form a single cluster in each case. However, in the AG and CG aquifers, also of different geological nature, the results related to the composition of the microbial communities placed those corresponding to the shallowest depths of both aquifers in the same cluster, while the samples representative of greater depths generated separate clusters. There are numerous studies that point to the complexity of the parameters which condition the composition of the microbiome of an aquifer [44], even considering the combination of deterministic and stochastic processes [84]. In this scenario, the salinity and the geological nature of the aquifer both appear to be nonprimary factors in terms of their contribution to microbial community shaping, as hypothesized by [85]. The authors of [86] suggested microbial communities are initially associated with the geological particularities of the aquifer, but are progressively differentiated according to external processes and disruptive events.

At the genus level, the differences between aquifers were accentuated. However, in all cases, except for in the PAL aquifer, a high variability of the bacterial community was observed, with high percentages of genera with relative abundances below 5%, generally increasing as depth increased (Figure 3B). This fact, already reported in other studies [25,87], supports the hypothesis put forward by [88], according to which a large percentage of the microbiota present in groundwater can be considered as transitional or part of the so-called "rare biosphere" [14,44,89]. The observed richness could be also influenced by the transport

of allochthonous taxa from the soils, and this connectivity with the surface will likely vary depending on whether the aquifer is karstic or detrital [90]. The high diversity involved in this distribution of the prokaryotic population gives the habitat a high plasticity that allows it to maintain the functionality of the system in a wide range of conditions. This plasticity can be particularly significant in situations such as the present one, wherein environmental conditions within aquifers are subject to great variation [78]. From another perspective, but also in terms of plasticity and resilience, these relatively high levels of diversity, with a high number of microorganisms showing low relative abundances, may be associated, in part, with the presence of populations that are poorly active or even metabolically inactive, typical of oligotrophic environments, as is the case in groundwater [91]. The fact that the percentage of species with relative abundances below 5% is considerably lower in the PAL aquifer, which, together with its low diversity, seems to confirm this hypothesis (according to which a pristine aquifer such as this one would show a much more stable and homogeneous microbial community), as already described by [92]. *Pseudomonas* and *Alcanivorax* were the genera to which the dominant OTUs in the PAL aquifer were assigned (Figure 3B). In the case of *Pseudomonas*, the metabolic plasticity it exhibits [93] allows it to be present in uncontaminated habitats, but with poor nutritional conditions unfavorable for the development of microorganisms with high nutritional requirements [94], as was observed. *Alcanivorax*, on the other hand, is more related to marine habitats, and its presence in groundwater is mostly attributed to waters with conditions that resemble seawater. The presence of *Roseovarius*, another genus with similar characteristics in this respect [95], seems to support this observation. However, this does not necessarily imply excessive salinity levels, since *Alcanivorax* has shown an ability to grow in a wide range of values, including the absolute absence of salt [96].

*Bacillus* and *Planococcus* were the dominant genera in the CG aquifer (Figure 3B). Several species assigned to both genera have been associated with environments exhibiting the presence of chemical contaminants [97–99]. The CG and AG aquifers present the highest concentrations of nitrates, above 100 mg/L (Table 1), linked to the high anthropic activity in this area. Both in the genus *Planococcus* and in *Bacillus*, numerous species tolerant to these higher salinity conditions have been described in the literature [100,101]. *Psychrobacter* also shares similar characteristics with respect to saline environments [102], so its presence in the samples from these aquifers seems to point to salinity as one of the factors responsible for the structure of its microbial community.

A similar population profile was detected in the shallower depth of the AG aquifer, where high concentrations of nitrates are also observed (30 mg/L, Table 1), with *Planococcus* as the dominant group. However, the increase in depth caused an important change in the composition of the microbiome, leaving only one OTUs as dominant, *FN436189* (Figure 3B), taxonomically related to the order *Solirubrobacterales*. The *Actinobacteria* assigned to this order are mostly related to soils and, although many aspects of their ecological significance are unknown, they are associated with environments exhibiting extreme conditions [103]. The existing difficulties to grow them in laboratory conditions greatly limit the available knowledge, despite which [104] postulate their cosmopolitan character, including aquatic environments based on their mobility and oligotrophic nature.

As has been observed in other Spanish coastal aquifers [21,22], the four aquifers studied here show great heterogeneity in their microbiological content. There is significantly greater inter-aquifer than intra-aquifer diversity, despite all of them being close to each other. The results obtained seem to outline that microbial communities are mostly conditioned by external factors, mainly anthropogenic in nature, with the role played by the physico-chemical properties of each aquifer being secondary, as has already been proposed by [52]. In this sense, the existence of high microbial diversity, with many low abundant taxa, could be indicative of anthropogenic influence on soil microbiomes, i.e., many of these microorganisms are not active, but merely remnants of human-impacted microbiota in groundwater. Furthermore, the higher diversity could result from a higher connectivity

with the surface, with little microbial connectivity between the aquifer and the sea, as has been reported in previous studies [22,105,106].

## 5. Conclusions

The coastal aquifers under study present a high heterogeneity in their microbial communities. Despite the considerable physico-chemical contrasts in the groundwater of the coastal aquifers, especially in the parameters related to salinity, the microbial communities appear to be most influenced by other external factors, such as anthropic effect due to the remarkable agricultural activity of this area. Compared to the characteristics of the pristine aquifer, where fewer of taxa exist, the microbiota in the altered media are characterized by a great diversity of genera, present in low proportions. This rare biosphere would have to have adapted to the new conditions brought about by the incorporation of pollutants into the underground medium, rather than occurring due to the nature of the aquifer itself, be it carbonate or detrital. As often detected in seawater, *Proteobacteria* is the dominant phylum in this type of media, with extreme environmental conditions; nevertheless, the presence of anthropic pollutants could promote the emergence of the phyla *Firmicutes* and *Actinobacteria*.

**Author Contributions:** Conceptualization: F.S. and Á.V. Collection and determination of the samples: F.S. and Á.V. Data interpretation: M.d.C.V.-G., F.S. and Á.V. All authors contributed to the written manuscript. Preparation of figures and tables: M.d.C.V.-G. and Á.V. All authors have read and agreed to the published version of the manuscript.

**Funding:** This research was supported by grants PID2019-108832GB-I00 from Ministry of Science and Innovation (MICINN) and UAL2020-RNM-B1953, co-founded by the European Regional Development Fund (ERDF) of the European Union (EU) and the University of Almeria.

**Data Availability Statement:** The sequence data were deposited in the NCBI SRA under the BioProject ID PRJNA894943.

**Acknowledgments:** This work is part of the general research lines promoted by the CEIMAR Campus of International Excellence.

**Conflicts of Interest:** The authors declare no conflict of interest.

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
