# Peer review of "Comparative Study of Microbial Diversity in Different Coastal Aquifers: Determining Factors"

_water, doi:10.3390/w15071337_

Round 1

Reviewer 1 Report

Please see attached PDF.

Thank you

Author Response

The answers/comments made to reviewer 1 have been made in the attached file.

Reviewer 2 Report

The work is very interesting and important from the point of view of aquatic ecology. The work has been done reliably and credibly, and the results are presented clearly.  The authors propose using microbial diversity as an indicator of the intensity of anthropopression (here, the work is applicable).

I have no many comments, and a few below:

The abstract requires additions in terms of taxonomic names (bacteria found) and numerical expressions (test results). As such, it has the character of a general outline.

Lina 25: Better anthropogenic impact than anthropic impact

Line 125: PVC - expand the abbreviation

Line 221: correct it: HCO3

State the conclusions in bullet points. In its present form, it presents the summarised results.

Author Response

All the comments have been made in the attached file.

Reviewer 3 Report

## Pg.6 - "Based on the physicochemical parameters measured, the Principal Component Analysis (Figure 2B)"

> please report data pre-processing step before PCA analysis (variable centring, scaling or normalization?)

## Pg.6 - "Similarly, the cluster analysis corroborates the importance of the salinity of groundwater (Figure 2C)"

> missing lots of information in order to explain this figure.

E.g.: Data from table 1 (pag.4) was pre-processed?  

      What was considered similarity metric in cluster analysis?  

      What was the Linkage algorithm?  

## Several undefined therms: OTUs, Q20, NCBI, BLAST, NBAYES, SIMPER, ...  

## pg.10 - in the legend of Fig.6

..."The Nearest Neighbour algorithm and Square Euclidean Distance were applied for hierarchical clustering of samples."

> "Nearest Neighbour" linking algorithm?  Are you sure?  I know as "single linkage"

> variables were pre-processed?

Author Response

Comments made in the attached file.

Round 2

Reviewer 1 Report

TheMS has been improved greatly. Still, I could see some statistical problems. Why you didn't use ANOVA? You must compare the differences between the samples. And, please add these results in your MS. Others are fine at this time.